# 3D Multilayered Metamaterials with High Plasmonic Hotspot Density for Surface—Enhanced Raman Spectroscopy

Jun Chen [1], Chai Zhang [2], Feng Tang [1] , Xin Ye [1,*], Yubin Zhang [1], Jingjun Wu [1], Kaixuan Wang [1], Ning Zhang [1] and Liming Yang [1,*]

1    Research Center of Laser Fusion, China Academy of Engineering Physics, Mianyang 621900, China
2    Joint Laboratory for Extreme Conditions Matter Properties, Southwest University of Science and Technology, Mianyang 621010, China
*    Correspondence: yexin@caep.cn (X.Y.); limy@ns.lzb.ac.cn (L.Y.)

**Abstract:** Three-dimensional (3D) plasmonic metamaterials have become a trend in the application of nanophotonic devices. In this paper, a convenient and inexpensive method for the design of 3D multilayer plasmonic metamaterials is constructed using a one-step self-shielded reactive-ion-etching process (OSRP) and a thermal evaporation system, which provides an efficient and low-cost method for the preparation of surface-enhanced Raman spectroscopy (SERS) substrates. The near-field enhancement of the 3D plasmonic metamaterials provides highly efficient electromagnetic resonance, and highly sensitive and uniform SERS sensing capabilities. The SERS detection results of rhodamine B (Rh. B) and rhodamine 6G (R6G) on this substrate show that the detection limit could reach $10^{-13}$ mol/L, and the signal could give expression to excellent uniform stability. The results show that high sensitivity and high robustness SERS substrates can be prepared with high efficiency and low cost.

**Keywords:** multilayer plasmonic metamaterials; one-step self-shielded reactive-ion-etching process; SERS substrates

## 1. Introduction

Up to now, metal nanostructures have received a lot of attention due to plasmonic resonance from collective oscillations of electron gases. When plasmonic resonance is excited, strong local electromagnetic field appears. The absorption and scattering of light beams are greatly enhanced by the nanostructures [1–5]. Based on these traits, 3D multilayered plasmonic metamaterials with high areal density and suitable nanogaps have been widely applied in photonic devices, i.e., photocatalysis [6,7], optical sensors [8–11], absorbers [12–15], and surface-enhanced Raman spectroscopy (SERS) [16–28].

Today, SERS has emerged as one of the most versatile tools for sensing and imaging chemical and biological analytes due to its high sensitivity and non-invasive advantages. The electromagnetic enhancement mechanism (EEM) has been widely recognized as a main enhancement mechanism of SERS. While some existing nanostructures have the potential to have extremely high SERS sensitivity, their practical application is often limited [21–25,27,28] because their electromagnetic enhancement "hot spots" have low density and poor uniformity. For example, Fang et al. chose Ag film on nanospheres (Ag/FON) as the SERS substrate [22]. At the "hot test" SERS-active point, the enhanced factor can reach $10^8$. However, these hot spots accounted for less than 0.1% of the total substrate and contributed 47% of the overall SERS intensity. However, in recent years, many 3D nanostructures have been applied to SERS, due to the fact 3D plasmonic metamaterials can provide a large number of evenly distributed hotspot densities and strong near-field electromagnetic fields. Garoli et al. prepared plasma metamaterials and proposed a test method, the gold filling factor f, estimated with a pixel count method. By modulating the fractal dimension of nanoporous gold, the effective dielectric response is customized over a wide spectral range of infrared wavelength [29]. Plasma metamaterials are used

in many ways because they provide powerful magnetic resonance and electromagnetic fields [30–36]. In the work of Liu et al., suspended 3D Ag nanoparticles/carbon nanotubes (Ag-NPs/CNT) nanohybrids for a SERS substrate are fabricated in high EEM "hots pots" by self-aggregating Ag-NPs onto the suspended CNT networks [23]. Zhang et al. proposed a hierarchical porous plasmonic metamaterial SERS substrate, and, using non-resonant benzenethiol as a probe molecule, the performance of the substrate is studied in detail [24], whose minimum detection limit is $10^{-12}$ M.

He et al. proposed an active substrate capable of super-sensitive detection of 10 fM Rhodamine 6G (R6G). In addition, the synthesized substrate can be applied to the marker-free detection of DNA with a sensitivity limit as low as 5 nM. The substrate is composed of a two-dimensional macroporous Ag film composed of a silver nanosheet (AgNS) -coated inverse opal film. However, the 3D plasmonic metamaterials manufacturing process has clear limits in that it is a complicated process, is suitable to high−throughput fabrication, and is high in process cost [32,35,36].

In this letter, a convenient and inexpensive 3D plasmonic metamaterials SERS substrate is proposed. The nano-protrusion-textured surfaces were obtained by OSRP on a fused-silica surface; after that, the silver layer, the silicon dioxide layer, and the gold particles cover the substrate in turn, forming 3D multilayered plasmonic metamaterials with high areal density. This 3D plasmonic nanostructure conforms to the industrial process and is convenient for mass production. These structures give rise to multiple near-field interactions between the top Au-NPs and the bottom Ag nanostructures as well as between the top Au-NPs themselves. The high density of hot spots in 3D space produces an efficient and widely tunable plasmonic response. This structure can be used as a new class of industrialized, highly efficient SERS substrates.

## 2. 3D Multilayered Plasmonic Metamaterials Design

The preparation process of 3D multilayered plasmonic metamaterials is shown in Figure 1, which is mainly divided into three steps: (1) Nano-protrusion-textured-by-etching-fused silica substrate is obtained [37]. (2) The substrate is coated with 40 nm Ag and 15 nm $SiO_2$ layers using a home-built thermal evaporation system. (3) Au-NPs are prepared and evenly rotated onto the substrate [38]. (4) The detection droplets are coated on the base and allowed to dry naturally for 30 h. The fused silica nano-protrusion-textured are obtained by fluorocarbon radical plasma etching on the stainless steel sample table of the RIE-3 system at an RF plasma frequency of 13.56 MHZ. Firstly, the chamber is cleaned with argon and oxygen plasma for 20 min and the fused silica substrate is put into the mixed solution (2:1 concentrated 65% $HNO_3$ and 30% $H_2O_2$) for ultrasonic cleaning. Trifluoromethane ($CHF_3$) and argon (Ar) plasmas are used in etching reactions, and the ratio of $CHF_3$ to argon is 10:50 SCCM. When the power is kept at 600 W and the pressure in the cavity is 5 Pa, the fused silica is etched for 25 min. Finally, we use alcohol, acetone, and a mixed solution (2:1 concentrated 65% $HNO_3$ and 30% $H_2O_2$) in turn to remove all of the impurities produced on the substrate during the etching process. The samples are coated with 40 nm thick of Ag and 15 nm thick of $SiO_2$ using the Nanguang ZZS900 film coater (Rankuum Machinery Ltd., Chengdu, China). Before coating, the material is first cleaned with an ultrasonic wave and then cleaned again with an oxygen ion beam. It is plated successively with 2 nm chromium, 40 nm Ag, and 15 nm $SiO_2$ film, where chromium film is an adhesive layer to enhance its structural robustness. The vacuum pressure is $5 \times 10^{-4}$ Pa, and the evaporation rate is 5A/s. Finally, the Au NPs stock solution with a diameter of 5 nm is prepared by the hydrothermal method and dripped onto the prepared substrate.

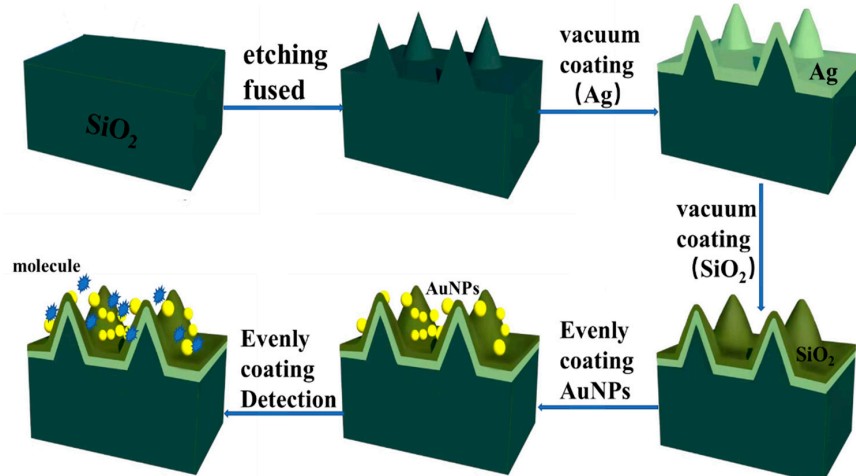

**Figure 1.** The preparation process of 3D multilayered plasmonic metamaterials with high areal density.

The 3D plasmonic metamaterials are clearly displayed using scanning electron microscopy (SEM) and atomic force microscopy (AFM). As shown in Figure 2a,b, the nano-protrusion-textured array has preferred vertical growth orientation. Nanostructures are randomly distributed and grow relatively uniformly. The uniform arrangement of the nano-protrusion-textured structures and the similar shape of each structure indicate the advantages of the OSRP in the large−scale preparation of microstructure arrays. The nano-protrusion-textured structures in Figure 2c,d is coated with 5 nm thick $TiO_2$ adhesive layer, 40 nm Ag film and 15 nm $SiO_2$ spacer layer on the surface, and 1 mL Au-NPs on the top. The structures are coated evenly, and the coating does not destroy the original characteristics of high density and high depth of the structure. The average period of the structure is 180 nm, and the average height is 270 nm. Au-NPs can be easily adsorbed to the coated nanotextured surface by electrostatic. After the addition of Au-NPs, the density and intensity of the near-field interaction of the structure are further improved. The narrow gap between Au-NPs and the $SiO_2$ spacer, as well as between Au-NPs, is more conducive to coupling to stimulate SERS activity.

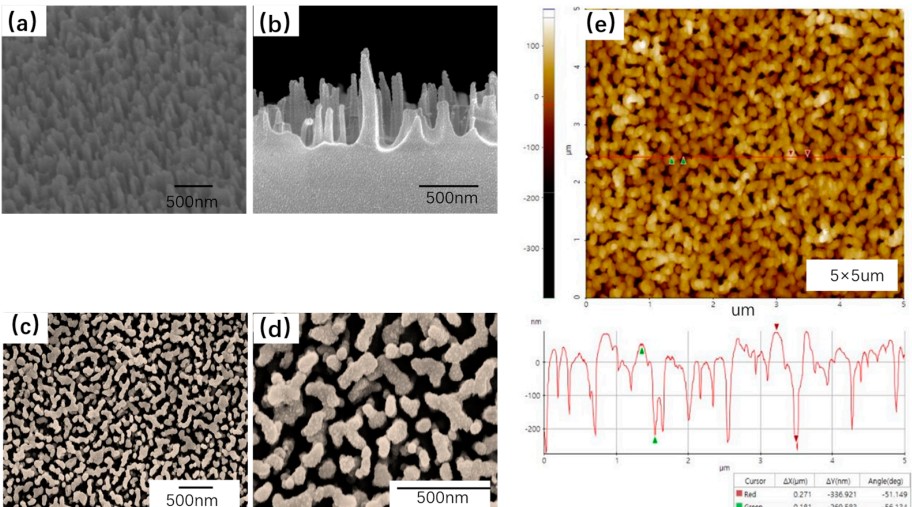

**Figure 2.** SEM of nano-protrusion-textured structure of fused quartz substrate without mask etch−ing: different magnifications (**a**) ×1000 nm, (**b**) ×500 nm. The nano-protrusion-textured is coated with a 5 nm thick $TiO_2$ adhesive layer, a 40 nm thick Ag film, and a 15 nm thick $SiO_2$ spacer layer, topped with 1 mL Au-NPs with a particle size of 5 nm different magnifications (**c**) ×1000 nm, (**d**) ×200 nm. (**e**) AFM of this structure and measurement of structural height and period.

## 3. SERS Performance

In order to reflect the excellent performance of 3D multilayered plasmonic metamaterials with multiple near-field interaction, we conducted SERS detection with it. Additionally, it is compared with the 2D plasmonic nanostructures. The electric distribution across plasmonic nanostructures is calculated using the finite-difference time-domain (FDTD) method. The incident electromagnetic field is set as a uniform plane wave at 522 nm, which propagates along the *z*-axis, and the polarization direction is the *x*-axis. Periodic boundary conditions were set around the nanostructure (*x*-axis and *y*-axis) with a period of P = 600 nm, and the upper and lower boundaries of the nanostructure were perfectly matched layers. The high strength and density electric field distribution on the 3D plasmonic nanostructure is realized by generating three near-field coupling effects: (1) between the top Au-NPs and the bottom $SiO_2$ nanostructure, (2) between Au-NPs on a single bottom nanostructure, and (3) between Au-NPs decorated on the sidewalls of two different adjacent structures. 3D multilayered plasmonic metamaterials compared with 2D structures are more conducive to arousing a strong local field in $SiO_2$ layer because in the direction parallel to the polarization direction of incident light and structure, it is not able to inspire the near-field coupling. However, the orientation effect for 3D nanostructures is very weak because the 3D nanostructures of the whole space there are always orientations perpendicular to the incident light polarization direction. The refractive index setting of the $SiO_2$ nanostructure, Ag layers, and Au-NPs is obtained from the literature [39,40]. The SEM and AFM of the structure as shown in Figure 2 show a uniform structure and strong robustness. However, there are tolerances for the radius and period of the structure. Moreover, the final structure shows a smooth columnar, and the selection of the columnar periodic structure in the simulation of the electromagnetic field. Three kinds of superstructures with different radii are simulated. The thicknesses of the Ag layer and the $SiO_2$ layer are constant, which are 40 nm and 15 nm, respectively; the columns of different radii affect the electromagnetic response of the overall structure, in Figure 3. Moreover, it is also reflected in the actual Raman detection. For example, the peak position and intensity of Raman measurement at different points of the substrate are biased. However, in terms of the overall effect, the non-uniformity of nanocrystals is limited, and the influence on the detection results is relatively small. Although the simulated structure is not completely similar to the actual structure, the comparison between the electromagnetic field intensity between one-dimensional, two-dimensional, and three-dimensional structure is still very valuable, which fully demonstrates that the three-dimensional structure can provide the powerful physical characteristics of the spatial electromagnetic field.

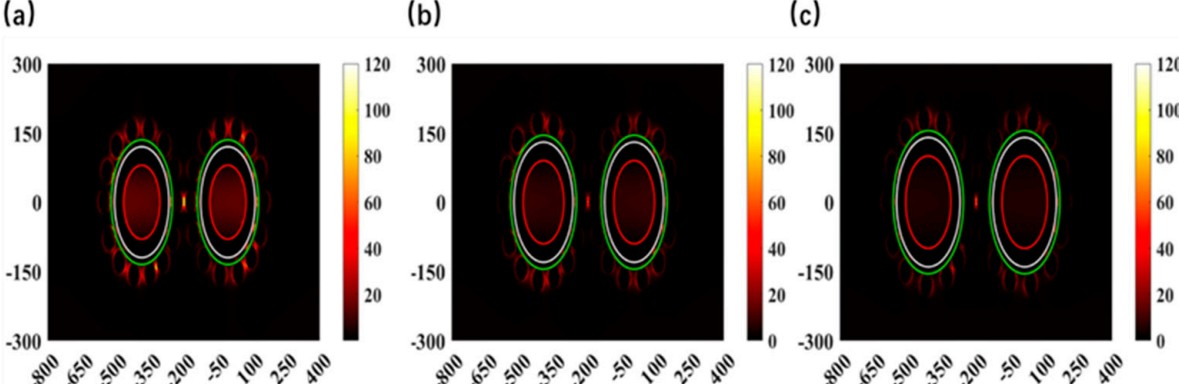

**Figure 3.** Electric field diagrams of nanostructures with different diameters. The thicknesses of Ag layer and $SiO_2$ layers are constant, which are 40 nm and 15 nm, respectively. The red, white, and green lines are $SiO_2$ column, Ag layer, and $SiO_2$ interval layer, respectively. (**a**) 160 nm; (**b**) 180 nm; and (**c**) 200 nm.

The Raman scattering process can be described as two processes. 1. Local field k local enhancement of the surface plasma effect excited Raman dipole 2. Radiation enhancement in Raman dipole radiation. The Raman enhancement factor can be expressed as

$$EF = EF_{Loc}(\omega_L) \, EF_{Rad}(\omega_R) \tag{1}$$

where $\omega_L$ and $\omega_R$ are the frequencies of the excitation light and radiation scattered light, respectively.

In practical calculations, it is often considered that $EF_{Loc}(\omega_L) \approx EF_{Rad}(\omega_R)$ and $\omega_L \approx \omega_R$. The average intensity of the SERS signal is approximately proportional to the fourth power of local electric field enhancement

$$EF = |E_{Loc}|^4 / |E_0|^4 \tag{2}$$

where $|E|$ and $|E_0|$ are the incident and local electric field intensity, respectively. In the simulation, all of the electric fields are normalized to the maximum electric field value of the incident TE fundamental mode. Thus, the Raman enhancement factor can be expressed as $EF = |E|^4$.

The electromagnetic simulation average *EF* of 3D multilayered plasmonic metamaterials is 5.6 times greater than that of 2D plasmonic nanostructures and 20.3 times greater than that of Au-NPs on smooth $SiO_2$ substrate, in the Figure 4a–c. As shown in Figure 4d, the SERS detection of $10^{-6}$ mol/L Rh. B is carried out by using three substrates under 522 nm excitation light. Interestingly, the Raman peak intensity is 1606.6 $cm^{-1}$, of 3D multilayered plasmonic metamaterials is about 52,799, of 2D plasmonic nanostructures is 9814, and of Au-NPs on smooth $SiO_2$ substrate is 3418. The Raman peak intensity is 1606.6 $cm^{-1}$, of 3D multilayered plasmonic metamaterials is about 5.38 times than that of 2D plasmonic nanostructures, and is about 15.5 times that of Au-NPs on smooth $SiO_2$ substrate. Intriguingly, the theoretical value is consistent with the measured SERS signal intensity. The results show that the 3D multilayered plasmonic metamaterials provide a superior structural design for generating strong near-field enhancement and high−sensitivity SERS substrates. Although the simulated structure is not completely similar to the actual structure, the comparison between the electromagnetic field intensity between the one-dimensional, two-dimensional, and three-dimensional structure is still very valuable, which fully demonstrates that the three-dimensional structure can provide powerful physical characteristics of the spatial electromagnetic field.

We compare our system performances with those of smooth glass substrate. This is the one way to prove that our approach is superior. As shown in the Figure 5, the R6G Raman peak on the smooth glass substrate does not have distinct characteristics, which may be the reason for the lack of enhancement effect. Another important reason may be the effect of fluorescence. Compared with the gold particle substrate on the glass substrate, the metamaterial substrate shows more distinct and strong Raman characteristic peaks. This is because 3D structures have a more uniform and robust spatial electromagnetic field (clearly shown in the electric field.), which can effectively enhance the Raman radiation.

The 3D multilayered plasmonic metamaterials induce multiple near-field interactions between the top Au-NPs and the underlying Ag layer; however, uniform interlayer thicknesses as well as uniformly distributed particle densities achieve uniform average near-field intensities. SERS detection is conducted on Rh. B with different concentrations (see Figure 6a. We selected five Raman peak sites, which are 1606.6 $cm^{-1}$, 1395 $cm^{-1}$, 1130 $cm^{-1}$, 856 $cm^{-1}$, and 376.4 $cm^{-1}$ respectively. Moreover, for trace detection at a low concentration of $10^{-13}$ mol/L, the characteristic peak position can still be displayed, reflecting the high sensitivity of 3D plasmonic nanostructures, which not only has high detection sensitivity but also has good uniformity and stability. This is a very low detection limit compared to other reports [33,41,42]. For example, the g−C3N4/Ag SERS substrate can be used to detect Rh. B with a linear relationship from $1.0 \times 10^{-9}$ to $1.0 \times 10^{-6}$ mol/L and a detection limit as low as 0.39 nmol/L [33]. The sodium salt of phytic acid (IP6)

stabilized Au@Ag core–shell bimetallic nanoparticles is used as SERS substrate. The limit of detection for RB in water is 5 nM (2 ppb) [41]. Kumar et al. prepared a SERS−active substrate composed of Au nanoparticles (NPs) on $Cu_2O$ microspheres. The corresponding limits of detection (LOD) are $2.36 \times 10^{-13}$ M for Rh. B and $3.40 \times 10^{-12}$ M for methylene blue (MB) [42]. In addition, the intensity of the spectral peaks at 1606.6 $cm^{-1}$ and 856 $cm^{-1}$ were chosen as the vertical coordinates, and the connection between the intensity and different concentrations was plotted in Figure 6b. To further reflect the substrate qualitative identification detection capability, another scheme (R6G) is chosen for SERS detection. As shown in Figure 6c, seven points in different parts are selected on 3D plasmonic metamaterial substrate (2 × 2 cm) to conduct SERS detection on a $10^{-10}$ mol/L R6G solution. Each measurement was separated by a week, and the SERS spectrum still showed a clear characteristic peak of R6G. However, the Raman strength decreases with the passage of time. Figure 6d displays the intensity variation with different times and places at 1593 $cm^{-1}$ and 613 $cm^{-1}$. It can be found that the intensity drops slightly at 613 $cm^{-1}$ and 1593 $cm^{-1}$; then the change is not significant. There are probably two main reasons. Firstly, it is known that SERS substrates usually lose their properties rather quickly due to uncontrolled deposition of carbon from the atmosphere. Therefore, maintaining the long−lasting activity of the sample is the key to the reuse of the base. In this paper, all sample storage is sealed with plastic wrap and placed in a drying oven. Secondly, the local strong electromagnetic field produced by different points on the substrate is different, and the long time may lead to the loss of detection molecules. There is no doubt that the 3D plasmonic metamaterials SERS substrate can be considered to have good reproducibility, which is mainly due to the fact that the nano−micro−pyramid array obtained through the OSRP and thermal evaporation system is quite uniform. However, the interlayer film thickness of the multilayer metamaterials and the Au-NPs on the surface cannot be absolutely uniform and homogeneous, which is an important factor preventing further reduction in the Raman intensity loss values. In addition, comparing the positions of the Raman characteristic peaks in Figure 6a,c, Rh. B and R6G, two substances with very similar chemical structures can be easily distinguished. The high robustness of using detection of this substrate is proved. In conclusion, because the SERS signal is averaged over the molecules adsorbed onto the plasmonic nanostructures, increasing the areal density of hot stops is the main contributor to enhancing signal uniformity. On the other hand, the $SiO_2$ spacing layer should also contribute to Raman signal uniformity in the 3D plasmonic metamaterials.

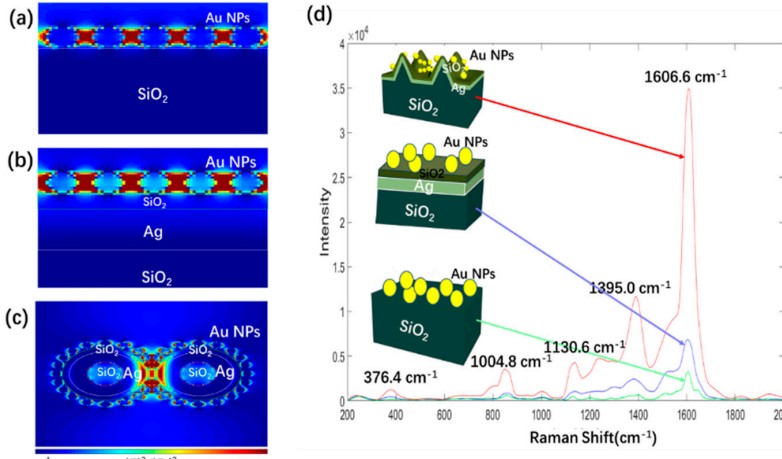

**Figure 4.** FDTD simulation of the electric field distributions, (**a**) Au-NPs decorated onto the smooth $SiO_2$ substrates, (**b**) 2D nanostructures, and (**c**) the 3D multilayered plasmonic metamaterials at the wavelength of 522 nm. The nanogap between Au-NPs remains constant in all three nanostructures. The experimental test, (**d**) SERS spectra measured from the $10^{-6}$ mol/L Rh.B treated 3D multilayered plasmonic metamaterials (red line), 2D plasmonic nanostructures (bule line), and Au-NPs on smooth $SiO_2$ substrate (green line).

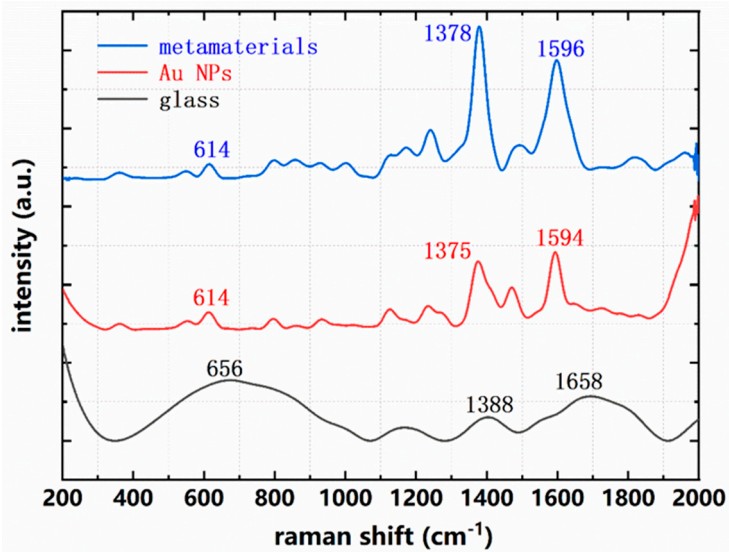

**Figure 5.** SERS spectra measured from the $10^{-8}$ mol/L R6G treated 3D multilayered plasmonic met ama terials (bule line), Au-NPs on smooth $SiO_2$ substrate (red line), and smooth glass substrate (black line).

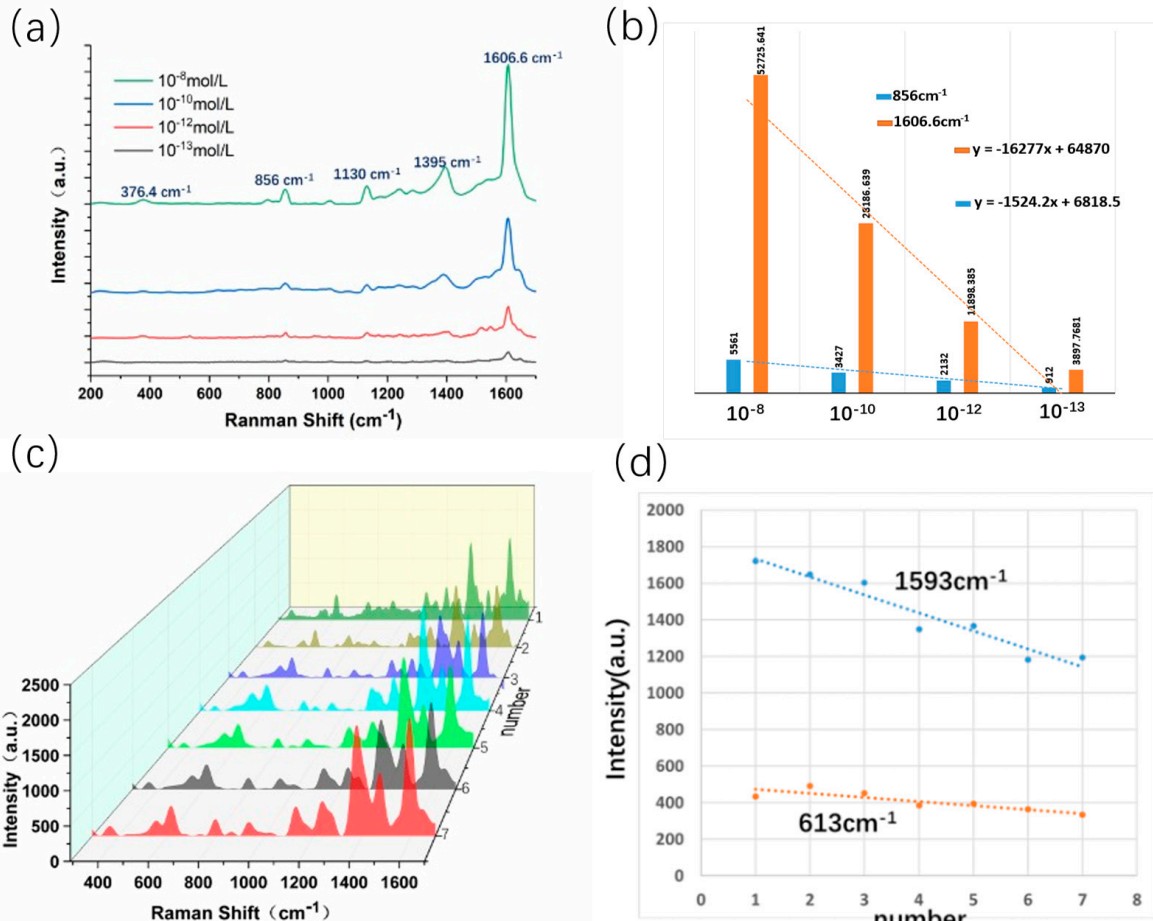

**Figure 6.** (**a**) SERS spectroscopy of Rh. B with different concentrations. (**b**) Intensity of Rh. B at 1606.6 cm$^{-1}$ and 856 cm$^{-1}$ as a function of different concentrations. (**c**) The characteristic spectra of $10^{-10}$ mol/L R6G solution are detected at the seven different points with seven measurements on $2 \times 2$ cm substrate (each measurement is separated by a week). (**d**) Intensity trend of R6G ($10^{-10}$ mol/L) at 1593 cm$^{-1}$ and 613 cm$^{-1}$.

## 4. Discussions

In summary, a highly sensitive and reproducible SERS substrate based on 3D plasmonic metamaterials substrate is constructed using the OSRP and thermal evaporation system, which provides an efficient and low−cost preparation process for a reproducible SERS substrate. The high strength and density near-field enhancement of the 3D plasmonic metamaterials provide highly efficient localized electromagnetic resonance fields, and highly sensitive and uniform SERS sensing capabilities. The average *EF* of 3D multilayered plasmonic metamaterials is 5.6 times that of 2D plasmonic nanostructures and 20.3 times than that of Au-NPs on smooth $SiO_2$ substrate. Using Rh. B and R6G as probe molecules, the performance of the 3D plasmonic metamaterials SERS substrate is studied in detail, whose minimum detection limit is $10^{-13}$ mol/L, and the signal gives expression to excellent uniform stability, indicating its high sensitivity. The manufacture of the structure not only meets the demand of convenient industrial mass production but also has excellent SERS capabilities.

**Author Contributions:** Conceptualization, J.C.; Methodology, J.C.; Software, J.C. and N.Z.; Validation, C.Z.; Formal analysis, X.Y. and N.Z.; Investigation, K.W.; Resources, C.Z. and Y.Z.; Data curation, J.W.; Writing—original draft, J.C.; Writing—review & editing, F.T.; Supervision, L.Y.; Project administration, X.Y.; Funding acquisition, L.Y. All authors have read and agreed to the published version of the manuscript.

**Funding:** This work was supported by the Innovation and Development Foundation of China Academy of Engineering Physics: CX20200021 and The Scientific Research Foud of Si Chuan Provincial Science and Technology Department: 2020YJ0137.

**Institutional Review Board Statement:** Not applicable.

**Informed Consent Statement:** Not applicable.

**Data Availability Statement:** Publicly available datasets were analyzed in this study. These data can be found here: [https://www.lumerical.com/], [RefractiveIndex.INFO—Refractive index database]. accessed on 1 October 2022.

**Conflicts of Interest:** The authors declare no conflict of interest.

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
