# Peer review of "3D Multilayered Metamaterials with High Plasmonic Hotspot Density for Surface—Enhanced Raman Spectroscopy"

_coatings, doi:10.3390/coatings13050844_

Round 1

Reviewer 1 Report

The manuscript describes a SERS plasmonic structure based on nano-protrusion-textured surfaces covered with the silver layer and silicon dioxide spacer, being finally capped with gold nanoparticles. The structure demonstrates good capabilities for detecting R6G and, as the authors report, is capable of performing not only qualitative but also quantitative measurements of the concentration of this analyte. The work may be of interest to specialists in the field of plasmonics, spectroscopy, nanotechnology, etc.

Manuscript remarks

 As the authors state: " … the linear functions between the intensity and different concentrations were plotted in Fig. 4(b). It is clear that between the two is approximately a straight line, proving the ability of the substrate to be quantitative detection". However, not without reason, in the paper “Chem. Sci., 2020, 11, 4563-4577" notes specifically that "despite of several works endorsing its quantitative capabilities, this characteristic is not well-recognized as a signature of SERS". Truly surface enhanced Raman scattering typically appears as a set of short flashes and random flickering dots on the surface of the SERS substrate. Moreover, the luminescence parameters of these points are affected by many changing factors (“burnout” of the analyte, local temperature, parameters of adhesion of molecules, etc.). Therefore, the identification of quantitative characteristics of SERS signals is a very non-trivial task that usually requires the accumulation of large statistical material and its thorough processing using adequate physicochemical and mathematical models. So, almost everyone who has dealt with SERS substrates shares the opinion that is also formulated in the work mentioned above: "The fingerprinting and ultrasensitive merits of SERS have positioned this technique as a powerful qualitative analytical tool." Of course, no one denies the fundamental possibility of obtaining a curve characterizing "the relationship between SERS intensity and analyte concentration". But the authors should explain in detail how they managed to solve this complex and non-trivial task and provide photos and videos demonstrating the “good uniformity and stability” of the SERS signal.

The manuscript states: "Each measurement was separated by a week, SERS spectrum still showed a clear characteristic peak of R6G." However, it is known that SERS substrates usually lose their properties rather quickly due to uncontrolled deposition of carbon from the atmosphere. Authors should explain why such aging is not observed in their case.

It is unclear how Figure 3c, illustrating an EM field simulation for a 3D structure, relates to the geometry of that structure. It is necessary to at least schematically show the location of the tips, the intermediate layers and the nanoparticle on the given map of the electromagnetic calculation.

Reviewer 2 Report

In their work, Chen and co-workers introduce a 3D multilayer plasmonic metamaterials fabricated using a one-step self-shielded reactive-ion-etching process and a thermal evaporation system, which provides a method for the preparation of surface-enhanced Raman spectroscopy (SERS) substrates. The near-field enhancement of the 3D plasmonic metamaterials, proved by using numerical simulations, provides highly efficient electromagnetic resonance, and highly sensitive and uniform SERS sensing capabilities. SERS detection results of rhodamine B (Rh. B) and rhodamine 6G (R6G) on this substrate shows that the detection limit could reach 10-13mol/L with a very good stability of the signal.

The article might be suitable for publication in a journal like Coatings, but before providing a final opinion, I would like to ask the Authors to address the following points.

1.      It’s not clear if the structure is porous-like. Compared to the ideal shape, the SEM images show that the pillars are arranged in a fractal-like geometry typical of porous materials. It would be important to estimate the degree of this fractal-like configuration since the Authors claim to have a metamaterial. A way to estimate is reported in literature (see ACS Photonics 2018, 5, 3408–3414). If this is the case, Authors should discuss potential further applications of their substrate in the context of porous materials (for instance for sequencing like in J. Phys. Chem. C 2020, 124, 41, 22663–22670, or in general for plasmonic applications, as recently reported by Garoli et al. in ACS Nano 2021, 15, 4, 6038–6060).

2.      Related to the previous point, why is this called metamaterial? The Authors should use effective medium to understand if they can describe this system with an effective medium theory. At that point, they can call it a metamaterial if the optical properties predicted by the EMT can reproduce the main optical features of the system simulated numerically.

3.      The simulated structures are ideal, from SEM images we have different heights (and slightly different diameters of the pillars), so I expect different effects.

4.      What is the real role of the pillars? It is not clear what it is gained having gold particles on this type of porous-like structure rather than on a flat substrate (the plasmonic behavior here comes from the nanoparticles, so I would expect they can do the job –typical plasmonic particle-enhanced SERS has been reported several times, for instance in Scientific Reports 2018, 8, 12652). In principle, the type of arrangement proposed by the Authors can enhance light-matter interactions and thus the SERS efficiency. I suggest the Authors to show at least one measurement where they compare their system performances with those of a flat substrate where gold particles are deposited. This is the only way to prove that their approach is superior.

5.      Pillar-like structures were used widely for SERS spectroscopy, and very important works should be discussed in the introduction or along the manuscript, also to make a fair comparison between the actual performances of this type of substrates with what has been already reported in literature (see for instance J. Mater. Chem., 2012, 22, 1370-1374, J. Mater. Chem. A, 2015, 3, 6408-6413, Nano Lett. 2013, 13, 11, 5039–5045, Nanoscale, 2016, 8, 11487-11493, Advanced Science 2018, 5, 1800560).

Reviewer 3 Report

This is well-written and well-documented contribution on an interesting topic. The work will certainly deserve publication after consideration of the following points.
1. In lines 142-143, authors indicate enhancements of 5.6 and 20.3 of the average SERS intensity. It is not clear whether these values come from the simulation or from the experiment: clarify, please. If they come from the simulation, please specify how the average values are calculated and how they connect to the simulation charts displayed in figure 3.

2. In lines 145-146, authors claim that the theoretical and measured SERS signal intensity are consistent. This is too vague for reader. Please, indicate the theoretical and the measured set of values, as well as the accuracy of these values.

3. In line 161-162, in conclusion and in abstract, authors highlight that a minimum detection limit of 10-13 mol/L is reached by using their system. Please, comment this value in comparison with the other systems presented in figure 3 or with systems reported in literature.

Round 2

Reviewer 1 Report

The revised manuscript looks much better than the previous version. I think that it can be published in its present form.

Reviewer 2 Report

The Authors have properly addressed my comments, so I support the publication of their work in Coatings.